# Interactions of EGFR/PTEN/mTOR-Pathway Activation and Estrogen Receptor Expression in Cervical Cancer

**DOI:** 10.3390/jpm13081186

**Published:** 2023-07-26

**Authors:** Thomas Bartl, Christoph Grimm, Robert M. Mader, Christoph Zielinski, Gerald Prager, Matthias Unseld, Merima Herac-Kornauth

**Affiliations:** 1Department of Obstetrics and Gynecology, Division of General Gynecology and Gynecologic Oncology, Medical University of Vienna, 1090 Wien, Austria; 2Department of Medicine I, Division of Oncology, Medical University of Vienna, 1090 Wien, Austria; 3Academy for Ageing Research, Haus der Barmherzigkeit, 1160 Wien, Austria; 4Department of Pathology, Medical University of Vienna, 1090 Wien, Austria

**Keywords:** cervical cancer, precision medicine, palliative treatment, mammalian target of rapamycin

## Abstract

(1) Objective: Late-line chemotherapy rechallenge in recurrent cervical cancer is associated with modest therapy response but significant side effects. As mTOR pathways modulate cellular growth via estrogen receptor (ER) signaling and combined mTOR and ER inhibition previously demonstrated survival benefits in breast cancer, this exploratory study evaluates mTOR pathway and ER expression interactions in a preclinical cervical cancer model. (2) Methods: Immunostaining of a 126-tumor core tissue microarray was performed to assess phosphorylated-mTOR and ER expression. To identify tumor subsets with different clinical behavior, expression results were matched with clinicopathologic patient characteristics, and both univariate and multivariable survival statistics were performed. (3) Results: phosphorylated-mTOR correlates with ER (r = 0.309, *p* < 0.001) and loss of PTEN expression (r = −2.09, *p* = 0.022) in tumor samples across stages but not in matched negative controls. Positive ER expression is observed significantly more often in phosphorylated-mTOR positive samples (30.0% vs. 6.3%, *p* = 0.001). In the subgroup of phosphorylated-mTOR positive tumors (n = 60), ER expression is associated with improved survival (*p* = 0.040). (4) Conclusion: ER expression appears closely intertwined with EGFR/PTEN/mTOR-pathway activation and seems to define a subgroup with clinically distinct behavior. Considering limited therapeutic options in recurrent cervical cancer, further validation of combined mTOR and ER inhibition in selected patients could appear promising.

## 1. Introduction

The therapeutic landscape of cervical cancer is rapidly evolving. The introduction of MRI-guided brachytherapy is currently revolutionizing first-line therapy in locally advanced cervical cancers, increasing 5-year disease-free survival up to 68%; in PD-L1 positive advanced or metastatic disease, the implementation of combined immune checkpoint inhibitor therapy with taxane-based cytotoxic chemotherapy increased 2-year overall survival to 53.0% and rapidly defined a new therapeutic standard [1,2]. Despite therapeutic advances, a fraction of patients experience cancer relapse and are referred to palliative care. 

Established palliative therapeutic options remain limited and are mostly confined to cytotoxic mono-chemotherapy. Late-line chemotherapy rechallenges, typically consisting of topoisomerase I inhibitors, gemcitabine, or vinorelbine, are associated with modest prognostic benefits but significant side effects. Tewari et al. reported mono-chemotherapy reinduction following progression after platinum-containing regimens to demonstrate response rates to be as low as 6.3% at an 8.5-month overall survival (OS); yet, 53.4% grade 3 events were observed, underlining the urgent unmet clinical need to identify new effective targeted approaches to improve palliative care [3,4]. 

In recent years, the EGFR/PTEN/mTOR pathway has received increasing research interest as a potential oncologic target in numerous solid tumors. Serving as a key intracellular signaling pathway of cellular growth, survival, and gene expression, the mTOR pathway is frequently hyperactivated during tumorigenesis [5]. Positive results of the RADIANT−3 study and the RECORD−1 study led to EMA-approvals of the mTOR-inhibitor everolimus for advanced renal cell carcinoma and metastasized neuroendocrine tumors of the pancreas [6,7]. In cervical cancer, underlying infections with high-risk human papillomavirus (HPV) provide a particularly strong rationale for mTOR inhibition, as the mTOR pathway appears to play a key role in virus/host cell crosstalk via oncoproteins E6 and E7. HPV infections are thereby associated with increased protein kinase B expression, which results in mTOR-pathway hyperactivation. Conclusively, in vitro mTOR inhibition leads to a decreased translation of E7 via 4E-BP1-protein phosphorylation blockade and subsequently induces cell cycle inhibition and apoptosis [5,8,9,10]. Early studies assessing mTOR inhibitor efficacy in heavily pretreated cervical cancer patients observed low objective response rates yet notably high rates of temporary disease control in up to 60% of patients with incurable disease, while mTOR inhibition appears relatively safe and well tolerated. The authors concluded that future studies testing combinations of mTOR inhibitors with other systemic treatment approaches would be of interest [11,12,13,14,15]. 

The mTOR pathway does not only crosslink with HPV-associated alterations of cell signaling but also modulates cellular growth by inducing ligand-independent estrogen receptor (ER) activation. mTOR pathway overactivation was therefore associated with clinical resistance to endocrine therapies. Consequently, combined mTOR and endocrine blockage demonstrated synergistic effects in a preclinical breast cancer model [16]. Based on these findings, the phase III randomized BOLERO−2 study assessed this rationale by comparing everolimus and exemestane versus exemestane and placebo in a randomly assigned 2:1 ratio in 724 hormone-receptor-positive advanced breast cancers, demonstrating a benefit in progression-free survival [17]. 

The therapeutic potential of combined mTOR and anti-endocrine therapy has previously been assessed for other hormone-receptor-positive solid cancers. In ER-positive endometrial adenocarcinoma, the mTOR pathway is altered in up to 80% of cases [18,19]. Whereas mTOR-inhibitor monotherapy approaches demonstrated low response rates, a study of combined everolimus and anastrozole reported a promising 32% overall response rate in 38 recurrent endometrial cancer patients after progression to platinum-based chemotherapy [20,21]. Response to mTOR inhibitor monotherapy was modest in recurrent ovarian cancers unselected for histology, mTOR mutational, and ER status, with response rates below 10% in two phase II trials [22,23,24]. While mTOR-pathway mutations are rare in high-grade serous ovarian cancer, they are observed in up to 20% of endometroid ovarian cancers and in up to 40% of clear-cell ovarian cancers, which could provide a more promising rationale for future research [25]. To our knowledge, however, no clinical data on combined anti-mTOR and anti-endocrine approaches are available for ovarian cancer to date. Interestingly, the concept of combined anti-mTOR and anti-endocrine therapy also demonstrated promising results in prostate cancer. Preclinical studies hint that almost all recurrent prostate cancers develop aberrant mTOR-pathway signaling [26]. One phase II trial combining everolimus and the antiandrogen bicalutamide observed a 75% serologic response and 28 months overall survival in castration-resistant prostate cancer [27]. 

For recurrent cervical cancer, palliative therapeutic options are limited, and the prognosis remains poor. As mTOR inhibition previously demonstrated therapeutic effects in cervical cancer, and cervical cancer expresses ER, investigating the rationale of the BOLERO−2 trial in cervical cancer could appear promising [28]. For cervical cancer, no evidence is available to date whether mTOR-pathway activation and ER expression are also associated, as previously observed in breast cancer. The present study is therefore set out to describe potential associations of mTOR and ER expression in a preclinical cervical cancer model.

## 2. Materials and Methods

The study was designed as an exploratory preclinical study assessing 168 tissue samples of histologically confirmed cervical cancer patients. One commercially available tissue microarray (TMA) set containing 168 tissue cores of 126 Caucasian patients diagnosed between January 2010 and September 2011 (Array catalog number HUteS168Su01-BX, 126 tumor cores including matched tumor-negative adjacent cervical stroma cores of 42 cases, respectively) including clinical baseline and survival data were purchased from BioCat GmbH (Heidelberg, Germany). The TMA set was arranged by taking a single tumor tissue specimen from preexisting paraffin-embedded tissue blocks per patient case with a total of 168 specimens per array at a core diameter of 1.5 mm and a section thickness of 4 µm. 

To assess potential associations between hormone receptor expression and mTOR-pathway activation, TMA-immunostaining was performed with an anti-estrogen receptor monoclonal antibody (Ventana, clone: 790−4324 SP1, Tuscon, Arizona, AZ, USA, ready to use), anti-progesterone receptor monoclonal antibody (source: Ventana, clone: 790−2223 1E2, Tuscon, Arizona, AZ, USA, ready to use), anti-EGFR monoclonal antibody (source: Ventana, clone: 790−2988, Tuscon, Arizona, ready to use), anti-phosphorylated mTOR (p-mTOR) monoclonal antibody (source: Cell Signaling, clone: 2976 49F9, dilution 1:50), and anti-PTEN monoclonal antibody (source: Abcam, clone: Ab32199, Cambridge, UK, dilution 1:50) according to the manufacturer specifications (Appendix A). Sections were scored individually by two observers, including a specialized pathologist, blinded to clinical parameters. An H-Score was calculated for all immunostaining results as the product of intensity thresholds at a maximum of 300 (negative, 0; weak, 1; moderate, 2; strong, 3; with an overall score of <5 negative; 5–100 weak, 101–200 moderate, 201–300 strong) as previously described to allow for semiquantitative analyses [29,30]. One patient case with matched negative core (n = 1/126, 0.79%; n = 2/168, 1.19%) was excluded prior to further analysis for a rare histologic tumor type, small cell neuroendocrine. 

Whereas the feasibility of p-mTOR immunostaining for cervical cancer has been demonstrated before, no evidence of meaningful expression cut-offs is available to date. For descriptive statistics, we, therefore, defined two patient cohorts broken down by positive versus negative (</≥5) p-mTOR expression [5]. In line, positive versus negative ER expression (</≥5) was applied as a cut-off for further analysis. 

Statistical analysis was performed using SPSS^®^ (IBM Corp. Released 2020. IBM SPSS Statistics for Windows, Version 27.0. IBM Corp., Armonk, NY, USA) for Windows. Categorical variables were described using percentages and medians, including interquartile ranges. To compare dependent variables, Student’s *t*-tests, chi-squared tests, and one-way ANOVA were applied, where appropriate. As a Kolmogorov–Smirnov test failed to demonstrate the normal distribution of immunostaining results, nonparametric tests were applied for all further comparisons of staining results. Correlations between immunostaining results were described by Spearman’s correlation coefficient. To account for the high proportion of early-stage cervical cancer samples (FIGO I 55.2%; n = 69), a subgroup analysis was performed by pooling all samples FIGO II-IV (44.8%; n = 56). Survival analyses with the endpoint of OS, calculated as the time between the timepoint of diagnosis and the last reported follow-up, were performed fitting Cox proportional hazard models both univariately and multivariately with a stepwise hierarchical selection approach at a significance level *p* < 0.05. For Cox models, immunostaining results (EGFR, PTEN, p-mTOR, ER) were calculated as continuous variables at 20 H-Score–point increments; PR was not included in the survival analysis, as there was only one PR expression sample in the overall cohort. Associations between ER expression and survival according to p-mTOR status were calculated by applying log-rank tests and graphically depicted by Kaplan–Meier curves. For all effect estimates, 95% confidence intervals were computed. Two-sided *p*-values < 0.05 were considered statistically significant. *p*-values serve only descriptive purposes; hence no multiplicity corrections were applied. The present study was approved by the Ethics Committee of the Medical University of Vienna (no.: 1182/2019) and conducted according to the principles expressed in the Declaration of Helsinki. 

## 3. Results

### 3.1. Descriptive Statistics

In total, 125 tumor tissue cores and 41 matched tumor-negative cervical stroma cores were evaluated in the final analysis. Clinical baseline characteristics broken down by p-mTOR expression levels are given in Table 1. At an expression cut-off ≥5, 48.0% (n = 60/125) stained p-mTOR-positive. No significant differences in available clinical pretreatment characteristics were observed between the cohorts.

In terms of patient survival, OS was significantly associated with FIGO stages with a median survival of 74.0 (68.0–80.0) months for stage I (n = 69), 59.0 (33.8–79.3) for stage II (n = 30), 48.0 (37.0–74.0) months for stage III (n = 23), and 24.0 (19.0- not available) for stage IV (n = 3) (*p* < 0.001). At a 60-month follow-up, 94.2% (n = 65/69) of FIGO I patients were alive as compared to 49.1% (n = 26/53) of FIGO II and III patients (*p* < 0.001). 

### 3.2. Comparison of Tumor Tissue and Adjacent Negative Tissue Controls

Tumor-negative adjacent cervical stroma cores were available for 30.0% (n = 18/60) of the p-mTOR-positive cohort for further analyses. In the overall cohort (n = 125), tumor tissue showed higher semiquantitative expression levels of EGFR (200 (105–280) vs. 120 (70–165), *p* < 0.001) but lower expression levels of p-mTOR (0 (0–10) vs. 40 (10–80), *p* < 0.001) and ER (0 (0–0) vs. 60 (15–180), *p* = 0.191). Differences for PTEN (90 (10–195) vs. 80 (13–120), *p* = 0.252) and PR expression (0 (0–0) vs. 0 (0–5), *p* < 0.001) remained insignificant. Results remained stable in a matched analysis of tumor samples with available adjacent negative tissue controls (n = 41), as given in Table 2. A subgroup analysis of only FIGO II-IV tumor tissue samples with available adjacent negative tissue controls (n = 26) also demonstrated lower p-mTOR (*p* = 0.048) and ER (*p* < 0.001) expression levels in tumor samples; differences did not reach statistical significance for EGFR (*p* = 0.087), PTEN (*p* = 0.737) and PR (*p* = 0.102).

### 3.3. p-mTOR Expression Correlates with EGFR Expression, ER Expression, and PTEN-Loss in Tumor Samples but Not in Matched Negative Controls

In the overall cohort (n = 125), EGFR expression demonstrates a positive correlation with PTEN expression (r = 0.232, *p* = 0.015) and a negative correlation with p-mTOR expression (r = −0.309, *p* < 0.001). Moreover, p-mTOR expression is positively correlated with ER expression (r = 0.309, *p* < 0.001) and PTEN-loss (r = −2.09, *p* = 0.022). The effect remained stable in the subgroup (n = 41) of tumor samples with available matched negative controls (EGFR and PTEN r = 0.246, *p* = 0.009, EGFR and p-mTOR r = −0.207, *p* = 0.207; p-mTOR and ER r = 0.226, *p* = 0.014). No respective correlations could be observed in the subgroup of tumor-negative cervical stroma cores. In the subgroup of FIGO II-IV tumor tissue samples with available adjacent negative tissue controls (n = 26), both EGFR and p-mTOR (r = −0.450, *p* < 0.001) as well as p-mTOR and ER (r = 0.426, *p* < 0.001) expression demonstrated significant correlations.

PR expression correlated with ER expression in the overall cohort (r = 0.500, *p* < 0.001). In the subgroup of FIGO II-IV tumor tissue samples, no case with PR expression was observed.

### 3.4. A Patient Subset with Both Positive p-mTOR and ER Expression Is Associated with Improved Overall Survival

In the overall cohort (n = 125), p-mTOR expression (HR 1.45, 95%CI [1.02–1.31] *p* = 0.047) was univariately prognostic for OS, whereas EGFR, PTEN, and ER expression failed to demonstrate statistical significance. p-mTOR expression could not retain its prognostic significance in multivariate analysis (Table 3). Positive ER expression, as defined at a cut-off >5, was observed significantly more frequently in the group of p-mTOR-positive patients as compared to p-mTOR-negative patients (18/60 = 30.0% versus 4/63 = 6.3%, *p* = 0.001). Positive ER expression was associated with significantly improved OS in the subgroup of p-mTOR positive patients (*p* = 0.040) but not in the subgroup of p-mTOR negative patients (*p* = 0.775) (Figure 1).

To account for potential confounding of early tumor stage on ER expression and survival, the percentage of FIGO I tumors according to ER expression was evaluated. In total, 54.8% (23/42) of ER-negative cases were FIGO I, and 66.7% (12/18) of ER-positive cases were FIGO I (OR 1.71, 95%CI [0.76–3.85] *p* = 0.253). 

## 4. Discussion

Expression of p-mTOR demonstrates a significant correlation with ER expression across tumor stages and is associated with loss of PTEN in a preclinical cervical cancer model. A subgroup of patients demonstrating both p-mTOR activation and positive ER expression is associated with improved overall survival. 

Present observations of ER and EGFR/PTEN/mTOR-pathway expression levels are in line as respectively reported in the literature. Previous studies observed ER expression in up to 31–37% of squamous cell cervical cancers and reported an association with favorable prognosis. Expression levels decrease with increasing FIGO stage as tumor cells appear to lose ER receptors during cancer progression and de-differentiation [28,31]. Den Boon et al. previously commented on this well-described phenomenon by arguing that ER signaling retains its effect in cervical cancer by a cancer-specific paracrine stroma-to-tumor signaling pathway. Thereby, despite an apparent loss of ER expression in the tumor tissue, estrogen may still act as a tumor growth signal [32].

Several preclinical studies suggest the clinical applicability of anti-endocrine drugs in cervical cancer [33,34,35]. One early clinical trial evaluated the efficacy of tamoxifen monotherapy in recurrent non-squamous cell cervical cancer, reporting a response rate of 11.1% in 3/27 patients with one complete and two partial responses [36]. However, tamoxifen may not be an ideal agent as it may act as an ER agonist rather than ER antagonist in cervical tissue [37]. Despite a strong rationale and promising early results, no sizeable clinical trials are available to date.

In regard to EGFR/PTEN/mTOR-pathway alterations, the literature consistently reports mTOR to be active in cervical cancer and to be a key pathway modulating HPV-associated cell signaling alterations [10]. In line with present results, literature reports positive p-mTOR immunostaining expression levels in around 50–60% of HPV-positive tumors. Data, however, vary throughout the literature due to heterogenous study populations, small sample sized, and different staining methods, complicating inter-study comparisons of results [38,39,40]. Evidence on the therapeutic efficacy of mTOR inhibition in recurrent cervical cancer patients is limited but appears to have the potential to temporarily halt disease progression, even in heavily pretreated stages, while inducing only mild to moderate side effects. 

In a phase I study of 74 patients with advanced gynecologic cancers, including 13 recurrent cervical cancers (median four prior treatment regimens), Moroney et al. assessed a combination of liposomal doxorubicin, bevacizumab, and temsirolimus. Two cervical cancer patients achieved partial response (15.4%) and one stable disease (7.7%); however, the study design does not allow to estimate the therapeutic efficacy of temsirolimus alone. The authors concluded that the combination was safe and well tolerated [14]. 

In a phase I dose-escalation study, including 41 patients with advanced gynecologic cancers, including six recurrent cervical cancers (median four prior treatment regimens), Piha-Paul evaluated a combination of bevacizumab and temsirolimus. Two cervical cancer patients achieved partial response (33.4%). The treatment combination was well tolerated and reversible by dose reduction or therapy pause [12]. 

A phase II study by Tinker et al. testing temsirolimus monotherapy in recurrent cervical cancer patients reported one partial response (3.0%) and 19 stable diseases (57.6%) in 33 patients available for response assessment, with a 6-month progression-free survival of 28% (14–43%). Side effects were mild and well manageable [13]. 

Of note, currently available studies assessing the therapeutic effects of mTOR inhibition included recurrent cervical cancer patients irrespective of potential mTOR-pathway alterations. The predictive value of molecular and/or immunohistochemical assessments of mTOR-pathway alterations on mTOR inhibitor response, therefore, remains to be elucidated. As several biomarkers, such as PIK3CA mutations, PTEN loss, or p-mTOR expression, have previously been proposed to select patients who are more likely to profit from mTOR inhibition, it could be hypothesized that patient selection according to future biomarkers may even improve response rates to mTOR inhibition as previously reported for cervical cancer. Only Tinker et al. retrospectively assessed potential influences of PTEN and PIK3CA immunostaining and PTEN promoter methylation status; however, respective biomarker results were only available for a fraction of patients and results, therefore, remained inconclusive (e.g., PTEN immunostaining results were available for only four patients) [15].

Interestingly, to our knowledge, this is the first study to describe associations between p-mTOR immunostaining expression and impaired survival in recurrent cervical cancer. Although this observation is well established for numerous other solid tumors, no data for recurrent cervical cancer are available to date. The present observation is supported by two studies on primary cervical cancer: Faried et al. reported increased p-mTOR expression to be associated with impaired survival in 25 advanced cervical cancers undergoing cisplatin-based neoadjuvant chemotherapy [39]. Meng et al. observed mTOR expression assessed by real-time PCR to be associated with higher FIGO-stage, tumor de-differentiation, and impaired 3-year OS in 125 cases [41]. 

The present study is also the first to describe potential associations between EGFR/PTEN/mTOR-pathway alterations and ER expression levels in cervical cancer. In line with the available literature, a relevant subgroup of cervical cancers appears to harbor EGFR/PTEN/mTOR-pathway alterations, which, as we demonstrated, correlate with ER expression. While cervical cancers otherwise seemingly tend to lose ER expression with increasing tumor stage, p-mTOR-expressing cervical cancers appear to be more likely to retain ER expression. As the present study was not set out to evaluate the potential clinical implications of this finding, further research is necessary to validate and interpret this observation. However, we may conclude that mTOR signaling and ER expression appear also closely intertwined in cervical cancer as previously described for other solid tumors. Moreover, ER expression could identify a subtype of cervical cancer with distinct clinical behavior. As the mTOR pathway modulates cellular growth via ER signaling and mTOR-pathway activation has been associated with resistance to endocrine therapies before, this observation could provide a rationale to consider the concept of combined mTOR-inhibitor and anti-endocrine therapy as tested in the BOLERO−2 trial also in advanced cervical cancers [17,42].

Of note, health-related quality of life is to be considered of increased relevance, especially in heavily pretreated cervical cancer patients subjected to palliative management. Ongoing research efforts to harness novel therapeutic approaches currently focus on highly effective intravenous agents, such as the immune checkpoint inhibitor cemiplimab or the antibody drug conjugate (ADC) tisotumab vedotin. Despite impressive outcomes in selected patients, intravenous application routes necessitate more time and resource-intensive hospital visits, and ADCs especially require close clinical monitoring to manage potentially fatal ocular adverse events [4,43]. As oncologic patients undergoing palliative care are reported to prefer oral treatment over intravenous application, repurposing well-established mTOR inhibitors and anti-endocrine agents for advanced cervical cancer could prove particularly beneficial [44]. Both drugs are available for oral administration and show toxicity profiles manageable in outpatient settings.

The present study faces limitations. First, its lack of random patient selection and possible erroneous data acquisition are to be considered typical for retrospective studies. Moreover, available clinical patient information did not include details on the type and extent of primary and/or recurrence therapy. As the present study was planned as purely exploratory to preclinically assess a potential therapeutic target of combined mTOR and ER inhibition, however, incomplete clinical details on specific treatment schemes do not impair the present study’s observations. To test the validity of available survival data, formal survival analysis broken down by the FIGO stage was performed. Results were in line with those previously reported by the EMBRACE study [2]. With respect to the specific research question of the present study, a clinically meaningful bias due to flawed therapy and/or survival data, therefore, appears unlikely. Second, matched negative controls allowing for direct comparisons of tumor and control tissues are not available for all patients. However, the proportion of available matched negative controls does not show statistically significant differences between the p-mTOR-positive and p-mTOR-negative tumor core groups. This shortcoming may therefore result in an underreporting due to impaired statistical power, but a relevant distortion of study results appears unlikely. Third, the available sample size only includes a total of 3 (2.4%) cases of adenocarcinomas. Present observations are, therefore, mainly limited to squamous cell cancer; conclusions regarding the clinically highly relevant subgroup of adenocarcinoma should be considered with caution. Fourth, 58.3% (n = 35) of all analyzed mTOR-positive tumor samples are derived from patients with FIGO I tumor stage. As any therapeutic approach, including mTOR inhibitors in recurrent cervical cancer, would consider patients with recurrent disease for whom no standard therapy is available anymore, respective tissue samples may not perfectly depict the molecular and clinical behavior of very advanced cancers. Due to the limited sample size of the cohort of interest, the performance of meaningful multivariable analyses was not feasible; therefore, we performed subgroup analyses for FIGO II-IV stages only to account for this shortcoming, which did not demonstrate significant differences in early-stage cancers. In summary, despite anticipating potential data-related flaws and minimizing the risk of bias, it is to be considered that the present study was planned as purely exploratory. As this is the first report on the mTOR pathway and ER intertwinement in cervical cancer, our findings may thereby offer interesting hypotheses for further validation.

## 5. Conclusions

EGFR/PTEN/mTOR-pathway activation and ER expression appear closely intertwined in cervical cancer, as previously observed in other solid tumors. p-mTOR-expression-positive cervical cancers seem more likely to retain ER expression, and expression of combined features seems to describe a subset of cervical cancers with distinct clinical behavior. As therapeutic alternatives for recurrent cervical cancer patients are limited, further evaluation of combined mTOR and ER targeting in selected cervical cancer patients could appear promising to improve palliative care.

## Figures and Tables

**Figure 1 jpm-13-01186-f001:**
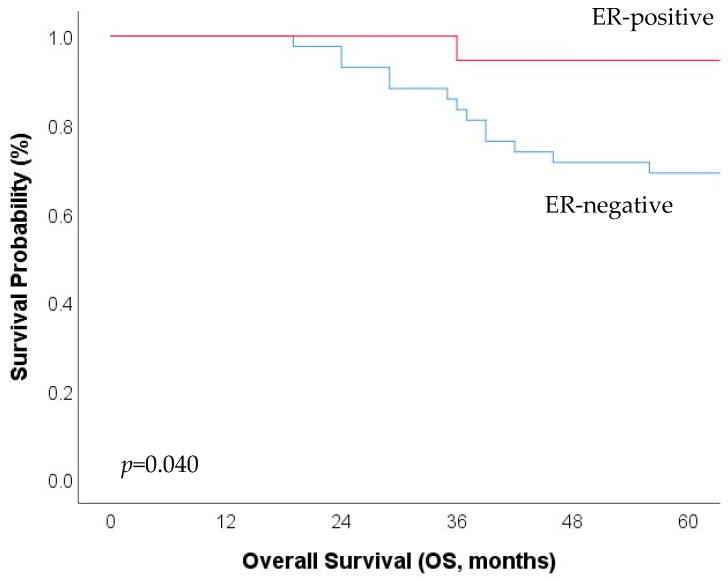
In the subgroup of p-mTOR-positive cervical cancer patients, positive estrogen receptor (ER) expression (H-Score ≥ 1) is associated with significantly improved overall survival (OS). ER-positive patients are marked in red (n = 18) and ER-negative patients are marked in blue (n = 42).

**Table 1 jpm-13-01186-t001:** Descriptive characteristics of pretherapeutic clinical baseline characteristics of all patients included in further analysis, broken down by positive or negative p-mTOR immunostaining result of respective tumor cores. Values are given as median (interquartile range) or number (%). p-mTOR staining remained inconclusive in two cases.

Parameter	All Patients	p-mTOR-Positive	p-mTOR-Negative	*p*-Value
number of patients	125 (100%)	60 (48.0%)	63 (50.4%)	0.664 *
controls available	41 (32.8%)	18 (30.0%)	22 (34.9%)	0.571 ^†^
age at diagnosis (years)	46.0 (41.0–55.0)	46.0 (40.5–51.8)	46.0 (41.0–55.0)	0.974 *
histologic subtype				0.641 ^§^
squamous	116 (92.8%)	55(91.7%)	59 (93.7%)	
adenocarcinoma	3 (2.4%)	3 (5.0%)	0 (1.2%)	
adenosquamous	6 (4.8%)	2 (3.3%)	4 (6.3%)	
histologic grading				0.310 ^†^
G1	6 (4.8%)	3 (5.0%)	3 (4.8%)	
G2/G3	99 (79.2%)	42 (70.0%)	56 (88.8%)	
missing	20 (16.0%)	15 (25.0%)	4 (6.3%)	
FIGO-stage				0.176 ^§^
I	69 (55.2%)	35 (58.3%)	32 (50.8%)	
II	30 (24.0%)	16 (26.7%)	14 (22.2%)	
III	23 (18.4%)	8 (13.3%)	15 (23.8%)	
IV	3 (2.4%)	1 (1.7%)	2 (3.2%)	
nodal status				0.113 ^†^
positive	24 (19.2%)	8 (13.3)	16 (25.4%)	
negative	101 (80.8%)	52 (86.7)	47 (74.6%)	
overall survival (months)	71.0 (50.0–79.5)	71.5 (62.3–80.8)	71.0 (43.0–77.0)	0.186 *

^†^ chi-squared, ^§^ one-way ANNOVA, * Student’s *t*-test.

**Table 2 jpm-13-01186-t002:** Immunostaining results in the overall cohort of all tumor cores (n = 125) and the subgroup of cores with available matched negative tumor-negative adjacent tissue controls. Immunostaining expression levels are compared between tumor cores and respective negative controls (n = 41, respectively).

Parameter	All Tumor Samples	Tumor Tissue with Available Controls	Matched Negative Controls	*p*-Value
number of samples	125 (100%)	41 (32.8%)	41 (32.8%)	
EGFR expression	200 (105–280)	190 (110–250)	120 (70–165)	0.010 *
positive	104 (83.2%)	33 (80.5%)	36 (87.8%)	
negative	9 (7.2%)	8 (19.5%)	4 (9.8%)	
missing	12 (9.6%)	0 (0%)	1 (2.4%)	
PTEN expression	90 (10–195)	100 (30–160)	80 (13–120)	0.145 *
positive	93 (74.4%)	32 (78.0%)	38 (92.7%)	
negative	28 (22.4%)	7 (17.1%)	2 (4.9%)	
missing	4 (3.2%)	2 (4.9%)	1 (2.4%)	
p-mTOR expression	0 (0–10)	0 (0–10)	40 (10–80)	<0.001 *
positive	42 (33.6%)	11 (26.8%)	36 (87.8%)	
negative	81 (64.8%)	29 (70.7%)	5 (12.2%)	
missing	2 (1.6%)	1 (2.4%)	0 (0%)	
ER expression	0 (0–0)	0 (0–0)	60 (15–180)	<0.001 *
positive	24 (19.2%)	7 (17.1%)	37 (90.2%)	
negative	101 (80.8%)	34 (82.9%)	4 (9.8%)	
missing	0 (0.0%)	0 (0%)	0 (0.0%)	
PR expression	0 (0–0)	0 (0–0)	0 (0–5)	0.001 *
positive	0 (0%)	0 (0%)	7 (17.1%)	
negative	123 (98.4%)	40 (97.6%)	33 (80.5%)	
missing	2 (1.6%)	1 (2.4%)	1 (2.4%)	

* related-samples Wilcoxon signed rank test.

**Table 3 jpm-13-01186-t003:** Univariate and multivariate Cox-regression analysis of parameters prognostic for OS in the overall cohort. All immunostaining results were calculated as continuous variables at 20 H-sore point increments.

	Overall Survival (OS)			
Parameters	Univariate Analysis		Multivariable Analysis	
	*p*-Value	HR (95% CI)	*p*-Value	HR (95% CI)
patient age	<0.001	1.13 (1.09–1.17)	<0.001	1.12 (1.08–1.16)
nodal status (N0/N1)	<0.001	4.26 (2.18–8.36)	0.003	2.97 (1.46–6.02)
FIGO stage (I/II vs. III/IV)	<0.001	4.43 (2.27–8.64)	-	-
grading (G1 vs. G2/3)	0.889	1.04 (0.58–1.87)	-	-
EGFR	0.416	0.97 (0.90–1.04)	-	-
PTEN	0.758	1.01 (0.95–1.08)	-	-
p-mTOR	0.047	1.45 (1.02–1.31)	0.549	1.04 (0.91–1.20)
ER	0.400	1.07 (0.91–1.25)	-	-

## Data Availability

The data that support the findings of this study are available from the corresponding author, [TB], upon reasonable request.

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
