# Peer review of "Interactions of EGFR/PTEN/mTOR-Pathway Activation and Estrogen Receptor Expression in Cervical Cancer"

_jpm, 2023, doi:10.3390/jpm13081186_

Round 1

Reviewer 1 Report

This is an interesting manuscript that lays out the options and challenges for combination therapy for various stages of cervical cancer.

The review is solid and informative, and presents an argument for the concerted application of ER and mTOR modulation.  I don't have any concerns or salient comments regarding this concept, although I would be interested in hearing more about how this fits into the broader context of cervical cancer treatment, such as:

  a) how the proposed strategies might complement, coordinate with, or potentially replace radiation treatment, and

  b) what are the relative limitations of the different targeting schemes vis a vis the practicalities of drug delivery.

There are probably a few minor typos.  I didn't scan carefully for them, though I did notice:

  line 78:  "is limited" should be "are limited".

Author Response

We thank the reviewer for the time and effort to review our manuscript and are grateful for positive response. Please find our replies below. In case the reviewer considers any of the provided information as relevant for the manuscript, we may add a paragraph on the reviewer’s discretion.

This is an interesting manuscript that lays out the options and challenges for combination therapy for various stages of cervical cancer.

The review is solid and informative, and presents an argument for the concerted application of ER and mTOR modulation.  I don't have any concerns or salient comments regarding this concept, although I would be interested in hearing more about how this fits into the broader context of cervical cancer treatment, such as:

1. how the proposed strategies might complement, coordinate with, or potentially replace radiation treatment, and

In my personal opinion, testing combined anti-mTOR + anti-endocrine approaches in late-line cervical cancers would not compete with currently established therapies, as it would be more of interest to offer it to strictly palliative patients, for whom no other options are available anymore (especially as we would expect rather a prolonged stable disease at the cost of low- to moderate side effects than impressive response rates). In line, I don’t see either/or situations in regard to radiation treatment. Radiotherapy has well-defined indications in palliative care and is very effective in e.g., controlling pain in bone mets. So both concepts could rather complement each other. Interestingly, early data suggests potential synergistic effects of radiation during anti-mTOR treatment – selected patients could even profit from simultaneous approaches (Nassim et al, PLoS One 2013, doi: 10.1371/journal.pone.0065257).

2. what are the relative limitations of the different targeting schemes vis a vis the practicalities of drug delivery.

I hope I understand the question correctly –outweighing treatment choices and mode of drug delivery in late line stages is a very personal and dynamic decision, for which few general statement can be made. In line with the answer to query A, combined anti-mTOR + anti-endocrine approaches would not directly compete with other options of clinical routine. Therapeutic gold 2nd-line standard is combined taxane-based chemotherapy + the immune checkpoint inhibitor pembrolizumab (Colombo et al, NEJM 2021, 10.1056/NEJMoa2112435). After progression to the latter, therapy choices are limited, but taxane-based combined chemotherapy reinduction, monochemotheray or immune checkpoint inhibitor monotherapy (if not previously applied) are to be discussed with the patient. At this stage of the disease, combined anti-mTOR + anti-endocrine could be an alternative option, depending on the patient’s preference. In such a strictly palliative situation, I would outweigh the time to be spent at the clinic (e.g. for i.v. application on site), risk of side effects and chance of response as most important factors (e.g. in case of a long interval since last chemotherapy, a fit and compliant patient and good tolerability during previous chemo, I would rather offer chemotherapy reinduction rather than experimental anti-mTOR + anti-endocrine) – but this decision would be totally up to the patient. In line with literature and personal experience, most patients would prefer oral therapy options, provided adequate patient compliance.

Reviewer 2 Report

Bartl and colleagues have presented a well-structured article entitled “Interactions of EGFR/PTEN/mTOR pathway activation and estrogen receptor expression in cervical cancer”. The article is quite interesting and presented a new concept related to the activation status of mTOR and estrogen receptor expression in cervical cancer. I have only minor comments regarding this study:

1- It would be interesting if the authors added a new paragraph in the introduction section and present the roles of mTOR activation status in other hormone-related cancers.

2- The are several small paragraphs in the introduction that introduce the same point so they can be combined together.  

3- Lines (224-226) are difficult to understand, please rephrase. Additionally, there is a punctuation error in line 224, kindly edit that. 

4- Authors showed that this is the first study that correlates mTOR activation status and estrogen receptor expression in cervical cancer. Moreover, that pathway has been studied in other cancers such as breast cancer. It would be interesting if the authors looked for a similar result of a subgroup with activated mTOR with ER expression associated with improved survival. Is that a universal finding or only a cervical cancer-related observation?

Minor editing of the English language required

Author Response

We appreciate the reviewer’s time and effort and thank for the constructive review. Please find our answers down below. If there are any further comments or remarks, we’ll gladly further adept the manuscript.

Bartl and colleagues have presented a well-structured article entitled “Interactions of EGFR/PTEN/mTOR pathway activation and estrogen receptor expression in cervical cancer”. The article is quite interesting and presented a new concept related to the activation status of mTOR and estrogen receptor expression in cervical cancer. I have only minor comments regarding this study:

  1. It would be interesting if the authors added a new paragraph in the introduction section and present the roles of mTOR activation status in other hormone-related cancers.

We may thank the author for the valuable point. We added a paragraph to the introduction, which – in our opinion - helps to contextualize and strengthen the hypothesis of the present study.

“The therapeutic potential of combined mTOR and anti-endocrine therapy has previously been assessed for other hormone receptor positive solid cancers. In ER positive endometrial adenocarcinoma, the mTOR-pathway is altered in up to 80% of cases [1,2]. Whereas mTOR-inhibitor monotherapy approaches demonstrated low response rates, a study of combined everolimus and anastrozole reported a promising 32% overall response rate in 38 recurrent endometrial cancer patients after progression to platinum-based chemotherapy [3,4]. Response to mTOR inhibitor monotherapy was modest in recurrent ovarian cancers unselected for histology, mTOR mutational and ER status, with response rates below 10% in two phase II trials [5-7]. While mTOR pathway mutations are rare in high-grade serous ovarian cancer, they are observed in up to 20% of endometroid ovarian cancers and in up to 40% of clear cell ovarian cancers, which could provide a more promising rationale for future research [8]. To our knowledge, however, no clinical data on combined anti mTOR and anti-endocrine approaches are available to date for ovarian cancer. Interestingly, the concept of combined anti mTOR and anti-endocrine therapy also demonstrated promising results in prostate cancer. Preclinical studies hint that almost all recurrent prostate cancers develop aberrant mTOR pathway signaling [9]. One phase II trials combining everolimus and the antiandrogen bicalutamide observed a 75% serologic response and 28 months overall survival in castration-resistant prostate cancer [10]. “ (new lines 76-94)

  1. The are several small paragraphs in the introduction that introduce the same point so they can be combined together.  

We followed the suggestion and merged the paragraphs introducing the rationale for cervical mTOR- targeting in the introduction.

  1. Lines (224-226) are difficult to understand, please rephrase. Additionally, there is a punctuation error in line 224, kindly edit that.

We thank the reviewer for the remark. We adapted the following sentence to increase its comprehensibility:

“As observed in the present study, tumor-adjacent stroma retains ER receptors -unlike cervical cancer cells-, and estrogen may thereby still play a role in form of paracrine stroma-to-tumor signaling” (former 224-226)

Explanation: This sentence referenced Boon et al. [11], who previously observed that ER receptor activation in cervical cancer does not only work via direct hormone-receptor interaction on cancer cells, but that activated estrogen receptors of non-cancerous, tumor adjacent stroma may also induce tumor growth via paracrine signaling. In the present study, we observed the same phenomenon as previously reported, namely that tumors tend to express less ER in increasing stages (except those with higher mTOR activation), while matched adjacent stroma retains its ER receptors.

The paragraph was adapted as follows:

“ER expression levels decrease with increasing FIGO stage as tumor cells appear to lose ER expression during cancer progression and de-differentiation. Den Boon et al. previously com-mented this well-described phenomenon by arguing that ER signaling retains its effect in cervical cancer by a cancer-specific paracrine stroma-to-tumor signaling pathway. Thereby, despite an apparent loss of ER expression in the tumor tissue, estrogen may still act as a tumor growth signal.” (new lines 240-245)

  1. Authors showed that this is the first study that correlates mTOR activation status and estrogen receptor expression in cervical cancer. Moreover, that pathway has been studied in other cancers such as breast cancer. It would be interesting if the authors looked for a similar result of a subgroup with activated mTOR with ER expression associated with improved survival. Is that a universal finding or only a cervical cancer-related observation?

We thank the reviewer for raising this important question. After re-reviewing the literature, studies for above mentioned hormone-positive tumors (compare answer to remark 1, lines 76-94) tended to assess the implications of mTOR mutations on hormone receptor positive tumors (~overcoming endocrine resistance), but not the implications for hormone receptor expression in mTOR mutated tumors. Whereas anti-endocrine therapies are well established in breast, endometrial and prostate cancer, they never entered clinical routine for cervical cancers. Due to the different perspective, previous authors – to our knowledge – did not assess the prognosis impact of ER on subgroups with activated mTOR in breast, endometrial and prostate cancer. As ER expression, however, is associated with good prognosis in all mentioned tumors, we would consider our results not surprising, but (indirectly) in line with the other tumors. If considered necessary, we could add these thoughts to the manuscript on the reviewer’s discretion.

  1. Oza, A.M.; Elit, L.; Tsao, M.S.; Kamel-Reid, S.; Biagi, J.; Provencher, D.M.; Gotlieb, W.H.; Hoskins, P.J.; Ghatage, P.; Tonkin, K.S.; et al. Phase II study of temsirolimus in women with recurrent or metastatic endometrial cancer: a trial of the NCIC Clinical Trials Group. J Clin Oncol 2011, 29, 3278-3285, doi:10.1200/jco.2010.34.1578.
  2. Cheung, L.W.; Hennessy, B.T.; Li, J.; Yu, S.; Myers, A.P.; Djordjevic, B.; Lu, Y.; Stemke-Hale, K.; Dyer, M.D.; Zhang, F.; et al. High frequency of PIK3R1 and PIK3R2 mutations in endometrial cancer elucidates a novel mechanism for regulation of PTEN protein stability. Cancer Discov 2011, 1, 170-185, doi:10.1158/2159-8290.Cd-11-0039.
  3. Slomovitz, B.M.; Jiang, Y.; Yates, M.S.; Soliman, P.T.; Johnston, T.; Nowakowski, M.; Levenback, C.; Zhang, Q.; Ring, K.; Munsell, M.F.; et al. Phase II study of everolimus and letrozole in patients with recurrent endometrial carcinoma. J Clin Oncol 2015, 33, 930-936, doi:10.1200/jco.2014.58.3401.
  4. de Melo, A.C.; Paulino, E.; Garces Á, H. A Review of mTOR Pathway Inhibitors in Gynecologic Cancer. Oxid Med Cell Longev 2017, 2017, 4809751, doi:10.1155/2017/4809751.
  5. Behbakht, K.; Sill, M.W.; Darcy, K.M.; Rubin, S.C.; Mannel, R.S.; Waggoner, S.; Schilder, R.J.; Cai, K.Q.; Godwin, A.K.; Alpaugh, R.K. Phase II trial of the mTOR inhibitor, temsirolimus and evaluation of circulating tumor cells and tumor biomarkers in persistent and recurrent epithelial ovarian and primary peritoneal malignancies: a Gynecologic Oncology Group study. Gynecol Oncol 2011, 123, 19-26, doi:10.1016/j.ygyno.2011.06.022.
  6. Emons, G.; Kurzeder, C.; Schmalfeldt, B.; Neuser, P.; de Gregorio, N.; Pfisterer, J.; Park-Simon, T.W.; Mahner, S.; Schröder, W.; Lück, H.J.; et al. Temsirolimus in women with platinum-refractory/resistant ovarian cancer or advanced/recurrent endometrial carcinoma. A phase II study of the AGO-study group (AGO-GYN8). Gynecol Oncol 2016, 140, 450-456, doi:10.1016/j.ygyno.2015.12.025.
  7. van der Ploeg, P.; Uittenboogaard, A.; Thijs, A.M.J.; Westgeest, H.M.; Boere, I.A.; Lambrechts, S.; van de Stolpe, A.; Bekkers, R.L.M.; Piek, J.M.J. The effectiveness of monotherapy with PI3K/AKT/mTOR pathway inhibitors in ovarian cancer: A meta-analysis. Gynecologic Oncology 2021, 163, 433-444, doi:https://doi.org/10.1016/j.ygyno.2021.07.008.
  8. Gasparri, M.L.; Bardhi, E.; Ruscito, I.; Papadia, A.; Farooqi, A.A.; Marchetti, C.; Bogani, G.; Ceccacci, I.; Mueller, M.D.; Benedetti Panici, P. PI3K/AKT/mTOR Pathway in Ovarian Cancer Treatment: Are We on the Right Track? Geburtshilfe Frauenheilkd 2017, 77, 1095-1103, doi:10.1055/s-0043-118907.
  9. Taylor, B.S.; Schultz, N.; Hieronymus, H.; Gopalan, A.; Xiao, Y.; Carver, B.S.; Arora, V.K.; Kaushik, P.; Cerami, E.; Reva, B.; et al. Integrative Genomic Profiling of Human Prostate Cancer. Cancer Cell 2010, 18, 11-22, doi:10.1016/j.ccr.2010.05.026.
  10. Chow, H.; Ghosh, P.M.; deVere White, R.; Evans, C.P.; Dall'Era, M.A.; Yap, S.A.; Li, Y.; Beckett, L.A.; Lara Jr, P.N.; Pan, C.-X. A phase 2 clinical trial of everolimus plus bicalutamide for castration-resistant prostate cancer. Cancer 2016, 122, 1897-1904, doi:https://doi.org/10.1002/cncr.29927.
  11. den Boon, J.A.; Pyeon, D.; Wang, S.S.; Horswill, M.; Schiffman, M.; Sherman, M.; Zuna, R.E.; Wang, Z.; Hewitt, S.M.; Pearson, R.; et al. Molecular transitions from papillomavirus infection to cervical precancer and cancer: Role of stromal estrogen receptor signaling. Proc Natl Acad Sci U S A 2015, 112, E3255-3264, doi:10.1073/pnas.1509322112.